# Enhanced Photon-Pair Generation Based on Thin-Film Lithium Niobate Doubly Resonant Photonic Crystal Cavity

Jinmian Zhu [1,2], Fengli Liu [1,2], Fangheng Fu [1,2], Yuming Wei [1,2,*], Tiefeng Yang [1,2], Heyuan Guan [1,2] and Huihui Lu [1,2,*]

1. Guangdong Provincial Key Laboratory of Optical Fiber Sensing and Communications, Jinan University, Guangzhou 510632, China; zhujinmian@stu2021.jnu.edu.cn (J.Z.); liufengli@stu2021.jnu.edu.cn (F.L.); ffh12@stu2021.jnu.edu.cn (F.F.); yangtiefeng2022@jnu.edu.cn (T.Y.); ttguanheyuan@jnu.edu.cn (H.G.)
2. Key Laboratory of Optoelectronic Information and Sensing Technologies of Guangdong Higher Education Institutes, Jinan University, Guangzhou 510632, China
* Correspondence: weiym280@jnu.edu.cn (Y.W.); thuihuilu@jnu.edu.cn (H.L.)

**Abstract:** In this work, a doubly resonant photonic crystal (PhC) cavity is proposed to enhance second harmonic generation (SHG) efficiency and photon pair generation rate (PGR). Through the exploration of geometry parameters, a band-edge mode within the light cone is identified as the first harmonic (FH) mode, and a band-edge mode outside the light cone is designated as the second harmonic (SH). Subsequently, by increasing the layers of the core region, a heterostructure PhC cavity is designed. The results showcase a doubly resonant PhC cavity achieving a 133/W SHG efficiency and a photon pair generation rate of $3.7 \times 10^8$/s. The exceptional conversion efficiency is attributed to the high quality factors Q observed in the FH and SH modes with values of approximately 280,000 and 2100, respectively. The remarkably high Q factors compensate for nonlinear efficiency degradation caused by detuning, simultaneously making the manufacturing process easier and more feasible. This work is anticipated to provide valuable insights into efficient nonlinear conversion and photon pair generation rates.

**Keywords:** photon pair generation; lithium niobate; doubly resonant; photonic crystals





## 1. Introduction

Photon pair sources play a pivotal role in quantum communications [1], quantum teleportation [2] and quantum computing [3,4]. Typically, photon pairs are generated through nonlinear processes, such as spontaneous parametric down-conversion (SPDC) [5–7] and spontaneous four-wave mixing (SFWM) [6,8,9]. In addition, the generation of photon pairs can be entangled, as in the case of polarized light [5], angular momentum [10] and frequency [11], or spatially and temporally coherent [8,12]. The photon pair generation rate (PGR) serves as a vital metric for evaluating the performance of photon pair sources. In nonlinear processes, the PGR is primarily determined by the nonlinear conversion efficiency of the structure. Hence, continually improving the nonlinear conversion efficiency of devices stands as a longstanding and crucial objective in nonlinear research.

The enhancement of nonlinear conversion efficiency in devices is a multifaceted endeavor, addressing both materials and structure designs. On one hand, it involves the search for materials with high nonlinear coefficients. On the other hand, it necessitates a structural design that can enhance the optical field intensity and extend the length of the interaction between the light and the material. For second-order nonlinear processes, a significant enhancement of the nonlinear conversion efficiency can be achieved by simultaneously enhancing the resonances at both the first harmonic (FH) and second harmonic (SH) frequencies [13–16], known as a doubly resonant cavity structure. In addition to the requirement of the high-quality factors (Q factors) of the two cavity modes, the nonlinear

overlap factor, involving the spatial overlap of the fields and the frequency-matching of the modes, must be sufficiently large to ensure an efficient frequency of conversion process.

Doubly resonant structures have been proposed for the periodic dielectric mirrors cavity [17–19], photonics crystal meso-cavity [20] and the photonic grating slab [21], and have been experimentally demonstrated in $Al_xGa_{1-x}As/AlAs$ mirrors microcavities [22], GaAs crossed beam photonic crystal nanocavities [23], aluminum nitride micro-ring resonators [24,25], Au plasmonic nanoantenna [26], and plasmon-fiber cavities [27]. Nevertheless, achieving a high nonlinear conversion efficiency using doubly resonant lithium niobate (LN) remains challenging, despite its outstanding electro-optic and $\chi^2$ nonlinear properties. Recently, a theoretical design based on a bound state in the continuum (BIC) has offered a novel avenue for the development of doubly resonant cavities on photonic crystal slabs [28]. By adding six new small holes around the large hole, the FH and SH can be matched in the $LiNbO_3$ structure [16]. However, it may be difficult to process such small holes (10 nm order) in nanomanufacturing. Therefore, a doubly resonant design which is compatible with the current techniques used in the nanofabrication of lithium niobate and a competitive PGR performance have drawn enormous interest in the fields of nonlinear and quantum optics.

Normally, SHG can occur in a doubly photonic crystal structure with a hexagonal lattice that respects inversion symmetry through a process called non-centrosymmetric SHG; it can be broken in a hexagonal lattice by introducing structural asymmetry. The matching of the phase velocities ensures an efficient energy transfer between the fundamental and second harmonic waves, while the doubly resonant conditions enhance the interaction between the waves and facilitate the conversion of the fundamental frequency to the second harmonic frequency and the periodic modulation of the refractive index allows for control over the dispersion properties. By introducing structural asymmetry into the doubly photonic crystal, such as by modifying the shape, size or arrangement of the constituent elements, the inversion symmetry can be broken. When the inversion symmetry is broken, the second-order nonlinear susceptibility allows for the generation of the second harmonic. This process involves the interaction of two photons at the fundamental frequency, resulting in the emission of a photon at twice the frequency (second harmonic). Moreover, the doubly resonant PhC cavity design abandoned the commonly held notion of engineering photonic bandgaps at both FH and SH frequencies. Instead, at the SH frequency, a BIC of the PhC slab is engineered to provide out-of-plane confinement, and a heterostructure of a hexagonal lattice is introduced to ensure the in-plane confinement in the absence of a photonic bandgap. On the other hand, at the FH frequency, the confinement mechanism is the same as in a conventional singly resonant PhC cavity, i.e., total internal reflection for the out-of-plane confinement and a photonic bandgap for the in-plane confinement.

Here, we present the generation of photon pairs at the telecom wavelength via an LN doubly resonant heterostructure. To simplify the fabrication process, the number of holes in outer regions is reduced to six. By optimizing the geometry parameters, a second harmonic conversion efficiency of 133/W and a PGR of $3.7 \times 10^8$/s are obtained. The configuration is a perfect photonics crystal (PhC) on a slab with thickness d and refractive index n, consisting of a hexagonal pattern of radius r with lattice period a, as displayed in Figure 1a. A horizontally heterogeneous structure is configured by introducing PhC regions with different hole radii, as illustrated in Figure 1b. Even without a band gap, the modes in the core region of this structure still have a divergent quality factor and the Q of the heterostructure modes can be large, especially when the core size increases. Thanks to the enlarged Q factor of the PhC cavity with heterostructure modes and finite size, the proposed doubly resonant structure provides valuable insights into efficient nonlinear conversion and photon pair generation application.

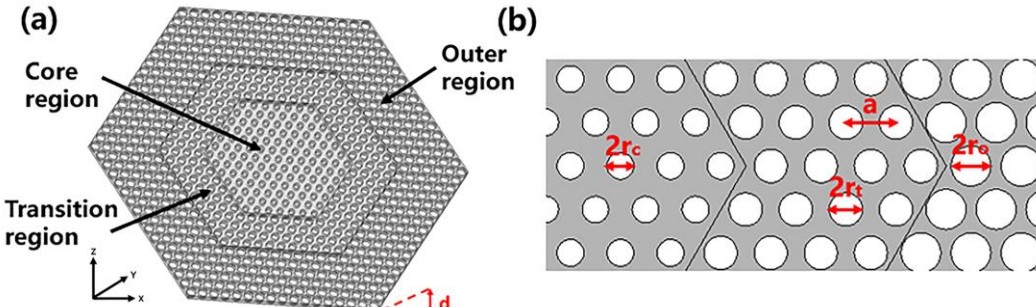

**Figure 1.** (**a**) Schematic diagram of lithium niobate photonic crystal slab of thickness d = 350 nm. (**b**) The heterostructure cavity with lattice period of a = 770 nm and air holes with radii of $r_c$ = 250 nm, $r_t$ = 260 nm, $r_o$ = 270 nm.

There are several reasons why a hexagonal lattice photonic crystal may be preferred over cubic or other lattice orders for second harmonic generation. Firstly, hexagonal lattice structures often exhibit favorable phase matching conditions for SHG. Phase matching refers to the condition where the wave vectors of the interacting waves (fundamental and second harmonic) match, allowing for efficient energy transfer. Hexagonal lattices can provide better phase matching compared to cubic or other lattice orders, resulting in a higher SHG efficiency. Secondly, hexagonal lattice structures can have specific crystal orientations or material properties that result in higher second-order nonlinear susceptibility. Nonlinear susceptibility determines the efficiency of SHG. By choosing a hexagonal lattice, one can optimize the crystal structure and material properties to enhance the nonlinear response and thus improve the SHG efficiency. Moreover, hexagonal lattices possess certain symmetry properties that can be advantageous for SHG. For example, hexagonal lattices can exhibit inversion symmetry, which allows for bulk contributions to SHG. This means that the nonlinear polarization arising from the second-order susceptibility can exist and contribute to SHG in the bulk of the crystal, leading to higher efficiency. Finally, hexagonal lattice photonic crystals offer more control over the dispersion properties compared to cubic or other lattice orders. By tailoring the lattice structure and the refractive index distribution, one can manipulate the dispersion characteristics, which can be crucial for achieving phase matching and optimizing SHG efficiency. As a consequence, the factors mentioned above make hexagonal lattice photonic crystals rather than cubic or other lattice orders, which is the preferred choice for SHG.

## 2. Model and Theory

The study was carried out by calculating the photonic structure bands of the LN PhC slab to satisfy the following design conditions through the following steps [28]:

1.  Calculate the dispersion relationship of photonic structure of the PhC using the finite element method (FEM, COMSOL Multiphysics 6.0).
2.  Identify a band-edge mode below the light line as the FH mode for frequency $\omega_1$, and a band-edge mode beyond the light cone at frequency $\omega_2$ as the SH mode.
3.  Verify that the FH and SH modes have a nonzero nonlinear overlap factor, depending on the $\chi^2$ tensor of the defined material.
4.  Vary the hole radius and the slab thickness to match the doubly resonant condition $\omega_2 \approx 2\omega_1$.
5.  Synthesize the FH and SH modes by appropriately configuring the PhC cavity.

We utilized the FEM to optimize the parameters d and r for the PhC to find modes that match the mentioned conditions. The dispersion characteristics of the medium are considered by using two different values of the refractive index of the slab for the quasi-TE and the quasi-TM simulations. For LN, we set n = 2.21 and n = 2.25 at wavelengths around 1550 nm and 775 nm, respectively. In Figure 2a, we show the photonic band structure for the quasi-TE modes of the PhC with d = 0.45a, r = 0.33a and n = 2.21; meanwhile, we also

show the quasi-TM modes of the same PhC but with n = 2.25 in Figure 2b. We can see that there is a pair of photonic bands that satisfy the required doubly resonant conditions, which is marked in green in Figure 2a,b.

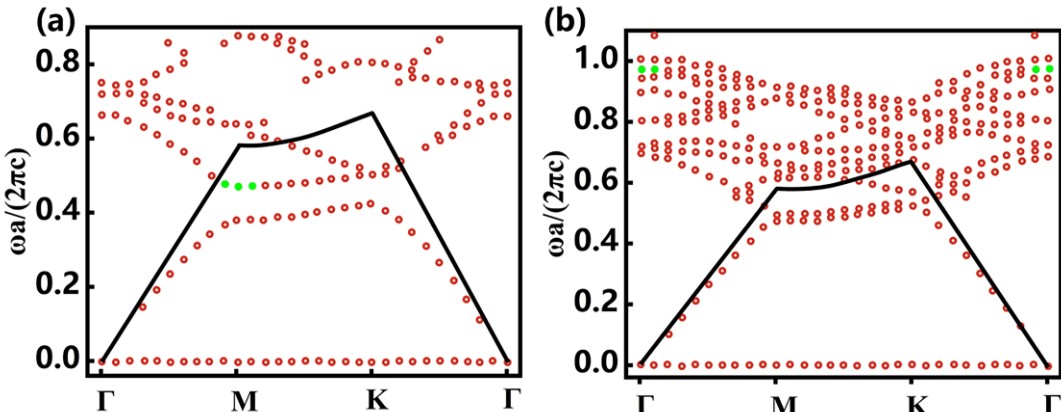

**Figure 2.** (**a**) Photonic band structure for quasi-TE modes of lithium niobate photonic crystal slab with d = 0.45a, r = 0.33a, index of refraction n = 2.21. (**b**) Photonic band structure for quasi-TM modes of lithium niobate photonic crystal slab with d = 0.45a, r = 0.33a, index of refraction n = 2.25. The green markers in (**a**,**b**) highlight the selected doubly resonant modes, the red markers are band structure.

Having identified the FH and SH modes, we now introduce the cavity mode through the heterogeneous structure design shown in Figure 1b. Three hexagonal regions were configured: core, transition, and outer, with radii of $r_c$, $r_t$, $r_o$, respectively. The region size is defined by the number of hexagonal "layers" of holes in each of them ($N_c$, $N_t$, $N_o$). For the resonant configuration that we studied, we set ($r_c$,$r_t$,$r_o$) = (0.33a,0.34a,0.35a). In this heterostructured configuration, the resonant mode can be confined vertically in the slab and laterally in the core region by the mode gap.

The resonant modes of the FH and SH were designed at wavelengths around 1550 nm and 775 nm, for which the design parameters are a = 770 nm, $N_c$ = 6, $N_t$ = 4, $N_o$ = 6, $r_c$ = 250 nm, $r_t$ = 260 nm, $r_o$ = 270 nm, d = 350 nm. Theoretical simulations were performed using the finite element method (COMSOL Multiphysics), in which the indexes of refraction used were n = 2.21 for FH and n = 2.25 for SH, respectively, to account for the dispersion characteristics of the medium.

For the doubly resonant modes in the photonic band structure shown in Figure 2a,b, it seems the modes have different in-plane momentums. Indeed, momentum mismatch can play a role in the process of second harmonic generation. In SHG, the conservation of momentum is an important factor. The momentum of the generated second harmonic wave should satisfy the momentum conservation law, which requires the total momentum of the interacting photons to be conserved. When there is a momentum mismatch between the fundamental wave and the second harmonic wave, the efficiency of SHG can be significantly affected. A perfect phase matching condition occurs when the wave vectors of the fundamental and second harmonic waves are exactly matched. In this case, the momentum conservation is satisfied, and the SHG process is highly efficient. However, if there is a momentum mismatch, the phase matching condition is not fulfilled, resulting in a reduced efficiency of SHG. The momentum mismatch can lead to a decrease in the conversion efficiency and may cause a spatial spreading of the generated second harmonic signal. Although the momentum mismatch between the modes plays a role in the SHG, a high quality factor and nonlinear overlapping factors help to compensate for the momentum mismatch and enhance the efficiency of SHG.

## 3. Results and Discussion

By using finite element simulations, we studied the relationship between the core size and the quality factor of the structure in the FH and SH. The outcomes are reported in

Figure 3. We can contest the fact that the theoretical Q factor of the structure builds up progressively as the size of the core region increases, and it can be inferred that as the size of the core region approaches infinity, the resonant mode will become a true BIC with a divergent quality factor.

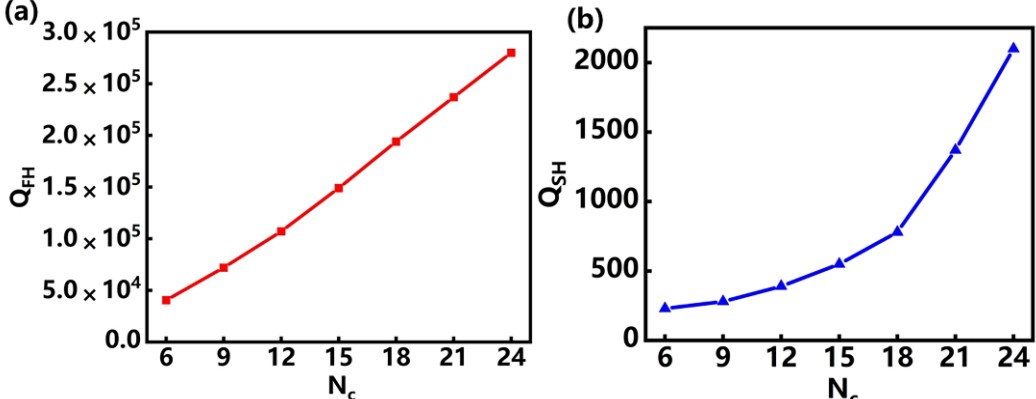

**Figure 3.** (**a**) The red line is the relationship between Q factor of the FH and core size. (**b**) The blue line is the relationship between Q factor of the SH and core size.

Here, we focus on the structures with $N_c = 24$, and it is possible to retrieve the electromagnetic field profiles inside the cavities from the simulations. As shown in Figure 4a–i, it is clear that the radiation is well confined to the core region in both the FH and SH. Both spatially and temporally confining the FH and SH modes is not a sufficient condition to ensure an efficient second-order nonlinear efficiency. The spatial overlap between the FH and SH fields must be suitably considered, as should their confinement volume. As for the second-order nonlinear processes, the overlap between the FH and SH resonant modes can be defined as [18]:

$$\overline{\beta} = \frac{\int d\mathbf{r} \sum\limits_{ijk} \overline{\chi}_{ijk} E_{2\omega i}^* E_{\omega j} E_{\omega k}}{\left(\int d\mathbf{r} \epsilon_\omega(\mathbf{r}) |\mathbf{E}_\omega|^2\right)\left(\int d\mathbf{r} \epsilon_{2\omega}(\mathbf{r}) |\mathbf{E}_{2\omega}|^2\right)^{1/2}} \lambda_{FH}^{3/2} \tag{1}$$

where $\lambda_{FH}$ is the wavelength in the free space in the FH; $\mathbf{E}_\omega$ and $\mathbf{E}_{2\omega}$ are the electric field profiles of the FH and SH, respectively; $\epsilon_\omega$ and $\epsilon_{2\omega}$ are the dielectric constants for the FH and SH, respectively; and $\overline{\chi}_{ijk}^{(2)}$ represents the elements of a dimensionless nonlinear tensor. In Figure 4j we demonstrate the relationship between the squares of the nonlinear overlapping factors and the core region size. The result indicates that as the core size increases, the nonlinear overlap factor becomes lower because the increasing cavity mode volume leads to a decrease in the nonlinear overlap factor. In the absence of external losses, the reduction in the conversion efficiency due to a decrease in the overlap factor can be fully compensated for by an increasing quality factor. This is because the increase in efficiency due to an increase in the quality factor is much greater than the decrease in efficiency due to a decrease in the overlap factor.

A sharp and strong peak can be observed in the wavelength range of the FH and SH (around 1550 nm and 775 nm), as illustrated in Figure 5a, as well as some higher-order peaks at small wavelengths. For the FH frequency, the principle of confinement is the same as in a traditional and single resonantly PhC, in other words, total internal reflection for the out-of-plane confinement and a photonic bandgap for the in-plane confinement [29]. In addition, for the SH frequency, a BIC of PhC is tailored to provide the out-of-plane confinement, and a heterostructured lattice with a hexagonal pattern is introduced to make sure the in-plane confinement in the absence of a photonic bandgap [30].

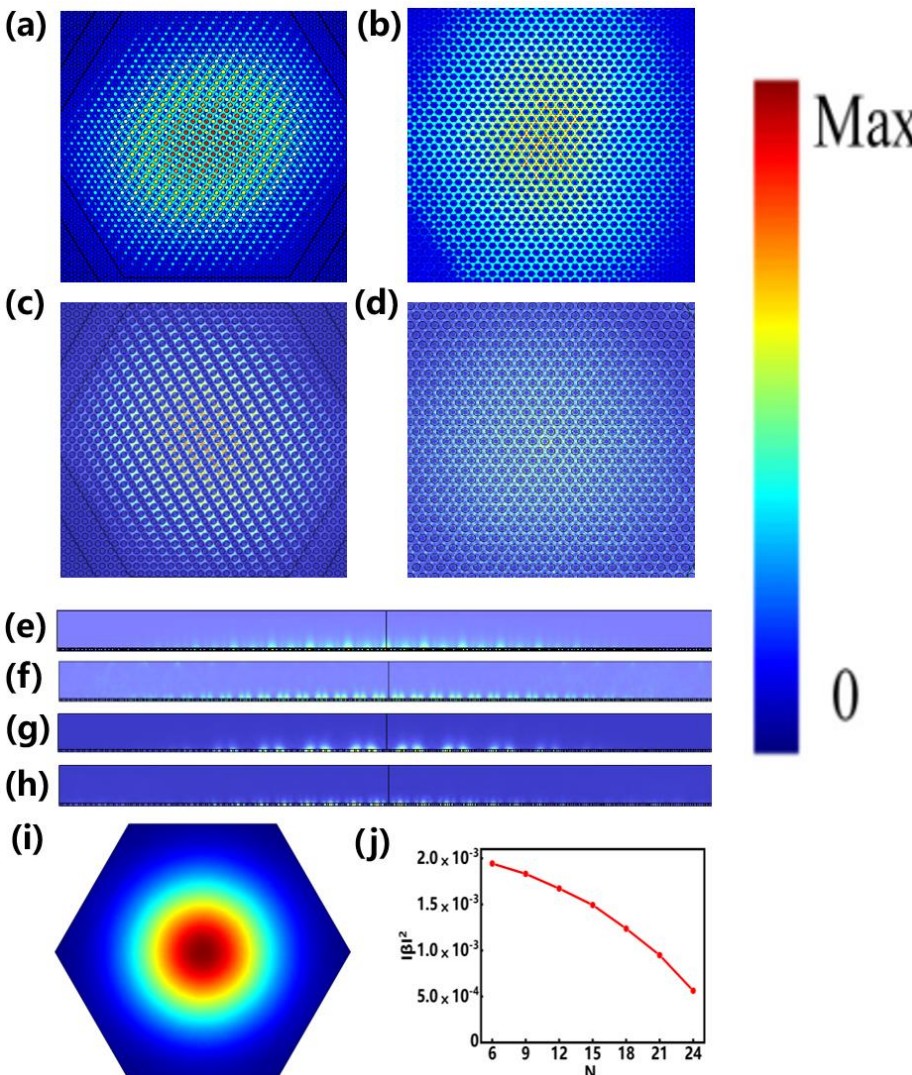

**Figure 4.** (**a**) Top view of the E field distribution of FH mode in the XOY plane at 1558.3 nm. (**b**) Top view of the E field distribution of SH mode in the XOY plane at 781.2 nm. (**c**) Top view of the H field distribution of FH mode in the XOY plane at 1558.3 nm. (**d**) Top view of the H field distribution of SH mode in the XOY plane at 781.2 nm. (**e**) Side view of the E field distribution of FH mode in the YOZ plane at 1558.3 nm. (**f**) Side view of the E field distribution of SH mode in the YOZ plane at 781.2 nm. (**g**) Side view of the H field distribution of FH mode in the YOZ plane at 1558.3 nm. (**h**) Side view of the H field distribution of SH mode in the YOZ plane at 781.2 nm. (**i**) Schematic diagram of the far-field mode in the XOY plane at 1558.3 nm. (**j**) The red line is the relationship between square of nonlinear overlapping factors and core region size.

In this part we discuss the wavelength shifts of the two resonances that influence the SHG conversion efficiency. The wavelength shift can be defined as [15]:

$$\Delta\lambda = \lambda_{FH}/2 - \lambda_{SH} \tag{2}$$

where $\lambda_{FH}$ and $\lambda_{SH}$ are the wavelengths of FH and SH, respectively. Furthermore, since the dependencies of the FH and SH resonant frequencies on the PhC parameters, such as the lattice constant a, the hole radius r, the size of the region and the slab thickness d, are different, tuning these parameters will lead to the detuning of the two resonances. As a matter of fact, the wavelengths of the resonant modes in the FH and SH are 1558.3 nm and 781.2 nm, as shown in the doubly resonant scattering spectrum in Figure 5a. Therefore, $\lambda_{FH}$ and $\lambda_{SH}$ in equation 2 are 1558.3 nm and 781.2 nm, so the value of $\Delta\lambda$ selected for

the calculation is 2.05 nm. For a PhC slab with an unchanged material and pumping condition, four parameters will affect the conversion efficiency: the resonant wavelength of the fundamental mode, the quality factor of the second harmonic mode, the quality factor of the fundamental mode and the factor of nonlinear overlapping. Although the wavelength shift of the two resonances leads to a reduction in the nonlinear conversion efficiency, the ultra-high quality factor possessed by the structure allows the efficiency reduction caused by detuning to be compensated for. The intrinsic conversion efficiency of SHG is calculated as follows:

$$\eta_{\text{conv}}(\lambda) \propto Q_{\text{FH}}^2 Q_{\text{SH}} \mathcal{L}_{\text{FH}}^2(\lambda) \mathcal{L}_{\text{SH}}\left(\frac{\lambda}{2}\right) \tag{3}$$

where $\lambda$ is the incident wavelength, $Q_{\text{FH}}$ and $Q_{\text{SH}}$ are the quality factors of FH and SH frequencies, $\mathcal{L}_{\text{FH}}(\lambda)$ and $\mathcal{L}_{\text{SH}}(\lambda)$ are the normalized intensities.

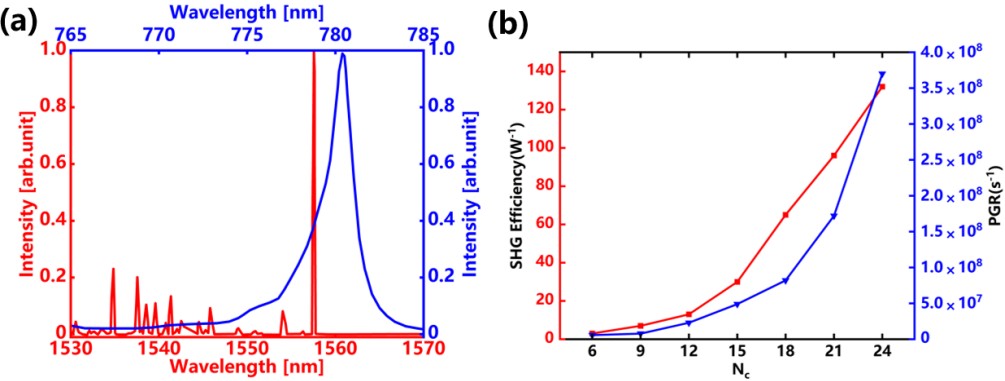

**Figure 5.** (**a**) The red line is doubly resonant scattering spectrum at FH and the blue line is at SH ($N_c$ = 24). (**b**) The red line is the relationship between core size and SHG Efficiency and the blue line is the relationship between core size and PGR.

As a function of wavelength, the FH and SH use the relation of Lorentzian function $1/\left(1 + (\lambda - \lambda_0)^2/(\Gamma/2)^2\right)$, where $\lambda_0$ is the resonant wavelength and $\Gamma = \lambda_0/Q$. When exciting the cavity at the FH resonance, $\mathcal{L}_{\text{SH}}(\lambda_{\text{FH}}/2)$ is 0.03. Therefore, in order to obtain a higher conversion efficiency, detuning should be minimized.

The performance of the proposed doubly resonant PhC cavity is then estimated. The SHG efficiency can be deduced by the following formula [21,28,31]:

$$\frac{P_o}{P_i^2} = \frac{8}{\omega_1} \left(\frac{\chi^{(2)}}{\sqrt{\epsilon_0 \lambda_{FH}^3}}\right)^2 |\bar{\beta}|^2 Q_{FH}^2 Q_{SH}. \tag{4}$$

where $\omega_1$ is fundamental frequency, and $\epsilon_0$ is the vacuum permittivity. Table 1 shows recent work on different kinds of doubly resonant photonic crystal cavities compared with our work. And the proposed doubly resonant photonic crystal cavity has a relatively larger SHG efficiency.

**Table 1.** Performance summary and comparison with state-of-the-art doubly resonant photonic crystal cavities.

| Reference | Material | $Q_{FH}$ | $Q_{SH}$ | $|\beta|^2$ | SHG Efficiency(/W) |
|-----------|----------|----------|----------|-------------|---------------------|
| [15] | GaN | 2000 | 800 | / | $2.4 \times 10^{-2}$ |
| [18] | AlGaAs | 5000 | 1000 | $1.6 \times 10^{-6}$ | 16 |
| [16] | LN | 160,000 | 2000 | $6 \times 10^{-4}$ | 48 |
| [31] | GaN | 36,000 | 1100 | $1 \times 10^{-4}$ | 2.9 |
| [31] | AlGaAs | 110,000 | 400 | $1.2 \times 10^{-5}$ | 112 |
| This work | LN | 280,000 | 2100 | $5.6 \times 10^{-4}$ | 133 |

The estimation of the PGR from the doubly resonant PhC cavity is conducted by evaluating the efficiency of the inverse process of SPDC, and namely sum frequency generation (SFG), both have the same origins due to quantum classical correspondence principle [32,33]. The total PGR across the region of the PhC cavity is then calculated using the following relation [33,34]:

$$\frac{1}{P_p}\frac{dN_{pair}}{dt} = 2\pi\eta^{SFG}\frac{\lambda_p^4}{\lambda_s^3\lambda_i^3}\frac{c\Delta\lambda}{\lambda_s^2} \tag{5}$$

where $P_p$ is the pump power of the light beam, which is 1 mW, $dN_{pair}/dt$ is the PGR per unit signal frequency and $\eta_{n_s n_i}^{SFG} = P_{SFG}/P_s P_i$ is the conversion efficiency of the sum frequency; $\lambda_p$, $\lambda_s$, $\lambda_i$ are the pump, signal and idler frequencies, respectively [35]. Here, we perform a direct nonlinear simulation of the process of photon pair generation in the doubly resonant structure using COMSOL Multiphysics, where the geometry of the model is a hexagonal environment, with a lithium niobate photonic crystal slab in the middle, and air on both sides of the photonic crystal slab. The polarization induced in the doubly resonant structure, which leads to SFG, was described with the LN $\chi^2$ tensor. The result is reported in Table 2, and we obtained a PGR of $3.7 \times 10^8$/s for $N_c$ = 24, which is significantly higher than the corresponding values for others. We also directly simulated the second harmonic conversion efficiency and photon pair generation rate for different core region sizes using finite element method (FEM, COMSOL Multiphysics), as shown in Figure 5b. We found that the SHG efficiency and PGR of the doubly resonant structure became higher as the core region size continued to increase.

**Table 2.** Summary of performance of PGR and comparison with others.

| Reference | Material | PGR (/s) |
|-----------|----------|----------|
| [36] | BBO | $1.02 \times 10^6$ |
| [37] | BBO | $3 \times 10^4$ |
| [35] | PPKTP | $1.6 \times 10^4$ |
| [34] | LN | $1 \times 10^4$ |
| [26] | Au | $3 \times 10^6$ |
| This work | LN | $3.7 \times 10^8$ |

As for the simulation of the photon pair generation process from the lithium niobate doubly resonant photonic crystal cavity, we calculate the photonic band structure of the PhC slab, the Q factor of the structure, the field profiles of the FH mode and the typical resonant scattering spectra at the FH in the linear regime. Spontaneous parametric down-conversion can be seen as the reverse process of sum frequency generation where two incident photons at frequencies $\omega_i$ and $\omega_s$ generate a photon at the sum of their frequencies $\omega_p = \omega_i + \omega_s$. For degenerate spontaneous parametric down-conversion, this reverse process is second harmonic generation. Therefore, we have nonlinearly simulated the second harmonic generation process of the inverse process of degenerate spontaneous parametric down-conversion by introducing the second-order nonlinear tensor of lithium niobate crystals, which leads to the field profiles of the SH mode, the typical resonant scattering spectra at the SH and nonlinear overlapping factors, as well as SHG efficiency and the PGR.

There are several reasons for this enhancement. Firstly, our doubly resonant photonic crystal cavity has larger dimensions compared to other structure. Secondly, our doubly resonant photonic crystal cavity possesses relatively pronounced resonances at both the pump and the decaying photon wavelengths. These resonances' wavelengths might improve the PGR via the enhanced electric field within the doubly resonant structure.

## 4. Conclusions

In conclusion, we simulated the enhancement of photon pair generation using an LN doubly resonant nanostructures and obtained a PGR of $3.7 \times 10^8$/s from an LN photonic

crystal slab with a thickness of 350 nm and an SHG conversion efficiency of 133/W. The photon pair generation is supported by FH and SH resonances, preceding the nonlinear efficiency as well as the photon generation rate of ordinary massive nonlinear crystals. A more efficient doubly resonant structure can be achieved by increasing the size of the core region, as discussed in this paper. How to balance the structure size to realize higher nonlinear efficiency and PGR needs to be considered comprehensively in further studies. The proposed doubly resonant structure based on thin-film lithium niobate PhC can pave the way for a solution for efficient nonlinear optics and photon pair generation devices.

**Author Contributions:** Conceptualization, J.Z. and H.L.; methodology, F.L.; software, F.F.; validation, Y.W., T.Y. and H.G.; formal analysis, H.L.; investigation, J.Z.; resources, F.L.; data curation, F.F.; writing—original draft preparation, J.Z.; writing—review and editing, H.L.; visualization, Y.W.; supervision, H.L.; project administration, J.Z.; funding acquisition, H.L. All authors have read and agreed to the published version of the manuscript.

**Funding:** This research was funded in part by the National Key Research and Development Program of China (2023YFA1407200), the NSAF (U2330113, U2030103, U2230111), the Youth Talent Support Programme of Guangdong Provincial Association for Science and Technology (SKXRC202304), the Natural Science Foundation of Guangdong Province (2023A0505050159, 2022A1515110970).

**Institutional Review Board Statement:** Not applicable.

**Informed Consent Statement:** Not applicable.

**Data Availability Statement:** Data will be made available on request.

**Conflicts of Interest:** The authors declare no conflicts of interest.

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
