# Peer review of "Enhanced Photon-Pair Generation Based on Thin-Film Lithium Niobate Doubly Resonant Photonic Crystal Cavity"

_photonics, doi:10.3390/photonics11050470_

Round 1

Reviewer 1 Report

Comments and Suggestions for Authors

The present manuscript is theoretical and numerical study on the development of a doubly resonant photonic crystal (PhC) cavity aimed at improving second harmonic generation (SHG) efficiency and photon-pair generation rate (PGR). The authors have conducted a detailed exploration of geometric parameters to enhance the resonant modes critical for SHG and PGR processes. The authors design a heterostructure PhC cavity that supports two band-edge resonances, where one is identified as the first harmonic (FH) mode within the light cone and the other is designated as the second harmonic (SH) outside the light cone. Due to their high-quality factors (approximately 280,000 for the FH mode and 2,100 for the SH mode), the simulated results indicate an SHG efficiency of 133/W and a PGR of 1.2×10^9/s.

This work will arise the interest of nonlinear nanophotonics, particular for the further step to some promising applications of LN-based photon-pair generation devices. The results are solid and convinced. I think the present manuscript has reached the quality of Photonics, so I would be happy to recommend publication in this version.

Reviewer 2 Report

Comments and Suggestions for Authors

The authors have calculated doubly resonant photonic crystal (PhC) cavity to enhance second harmonic generation (SHG) efficiency and photon-pair generation rate (PGR) with two selected band-edge modes. In general, this work does not have enough impact and innovation to reach the quality for publication. In addition, the following comments can be addressed.

The electromagnetic field distributions E and H of two cavity modes need to be plotted in both top and side views, in Fig. 4.

The wavelengths of the FH and SH at 1550 nm and 775 nm are used in the cavity design. From equation 2, Δ𝜆 is equal to 0.  So, in equation 6, what the value of Δ𝜆  is selected for the calculation.

The authors only simulated the photonic crystal cavity modes in linear regime. However, the photon-pair generation rate is just simply calculated by using equation 6.  The authors can consider to directly simulate the photon-pair generation process from the lithium niobate doubly resonant photonic crystal cavity.

Reviewer 3 Report

Comments and Suggestions for Authors

pleaes see attache documents for the review commets. 

Round 2

Reviewer 2 Report

Comments and Suggestions for Authors

This work does not have enough impact and innovation to reach the quality for publication. The photon-pair generation rate simply calculated by using equation 6 is around 1.2x10^9 /sec, which might be much higher than what can be achieved in reality. The authors may consider to directly simulate the photon-pair generation process from the lithium niobate doubly resonant photonic crystal cavity.
